# Ensemble-Instruct: Generating Instruction-Tuning Data with a Heterogeneous Mixture of LMs

**Young-Suk Lee, Md Arafat Sultan, Yousef El-Kurdi, Tahira Naseem**
**Asim Munawar, Radu Florian, Salim Roukos, Ramón Fernandez Astudillo**
{ysuklee,yousefelk,tnaseem,raduf,roukos}@us.ibm.com
{arafat.sultan,asim,ramon.astudillo}@ibm.com
IBM Research AI

## Abstract

Using in-context learning (ICL) for data generation, techniques such as Self-Instruct (Wang et al., 2023) or the follow-up Alpaca (Taori et al., 2023) can train strong conversational agents with only a small amount of human supervision. One limitation of these approaches is that they resort to very large language models (around 175B parameters) that are also proprietary and non-public. Here we explore the application of such techniques to language models that are much smaller (around 10B–40B parameters) and have permissive licenses. We find the Self-Instruct approach to be less effective at these sizes and propose new ICL methods that draw on two main ideas: (a) Categorization and simplification of the ICL templates to make prompt learning easier for the LM, and (b) Ensembling over multiple LM outputs to help select high-quality synthetic examples. Our algorithm leverages the 175 Self-Instruct seed tasks and employs separate pipelines for instructions that require an input and instructions that do not. Empirical investigations with different LMs show that: (1) Our proposed method yields higher-quality instruction tuning data than Self-Instruct, (2) It improves performances of both vanilla and instruction-tuned LMs by significant margins, and (3) Smaller instruction-tuned LMs generate more useful outputs than their larger un-tuned counterparts. Our codebase is available at `https://github.com/IBM/ensemble-instruct`.

## 1 Introduction

Instruction-tuned language models have demonstrated strong zero-shot generalization capabilities to new tasks (Chung et al., 2022a; Wei et al., 2021; Ouyang et al., 2022; Mishra et al., 2022; Wang et al., 2022; Longpre et al., 2023), creating interest in large-scale automatic synthesis of instruction-tuning data (Honovich et al., 2022; Wang et al., 2023; Xu et al., 2032; Sun et al., 2023a; Xu et al., 2023). In this context, Self-Instruct (Wang

et al., 2023) showed that a small number of expert-annotated seed examples, coupled with in-context learning (ICL) with a base model, can be used to generate an instruction-tuning dataset to efficiently instruct that same base model. While this method yielded strong results and multiple follow-up works, most techniques resort to very large LMs (around 175B parameters) (Wang et al., 2023; Taori et al., 2023), available only through closed-access APIs, or have restricted model access.

In this paper, we present Ensemble-Instruct, a novel algorithm enabling high-quality instruction-tuning data generation with smaller LMs (40B parameters or less), that are also fully accessible and have permissive usage licenses. We show that, when using smaller models as generators, Self-Instruct struggles to produce text of adequate quality, adversely affecting the utility of the generated data and downstream model performance. Staying within the ICL framework and using the Self-Instruct seed tasks, Ensemble-Instruct explores two main ideas to solve this problem: (1) Categorizing and simplifying the ICL prompts to ease the few-shot learning process, and (2) Ensembling over multiple LM outputs to improve both accuracy and diversity of the generated data.

A standard instruction-tuning sample exemplifies a task comprising: (a) an *instruction* that describes the action to be performed, (b) an optional *input* on which the action is performed, and (c) the *output* of the action. Similar to Self-Instruct, we generate samples in two stages: instruction generation and instance generation, where an *instance* comprises an input (optional) and an output. Unlike Self-Instruct, Ensemble-Instruct seeks to simplify the problem for the generating LM by first categorizing the examples into two types—those with an input and those without—and then employing separate pipelines for the two that leverage their own unique and simplified prompts (§2.1). Further, it ensembles over the outputs of different LMs in

two complementary ways: (1) including examples generated by a heterogeneous collection of LMs in the final set to increase diversity, and (2) majority voting followed by filtering low-consensus examples to improve accuracy (§2.4).

To understand the effects of our proposed methods, we run an extensive evaluation of different models for instruction generation. This includes vanilla language models (T5) UL2-20B (Tay et al., 2022), FALCON-40B (Penedo et al., 2023), the instruction-tuned models FLAN-T5-11B (Chung et al., 2022b) and FLAN-UL2-20B (Tay et al., 2022) and the chat-tuned[1] version of GPT-NeoX-20B (Black et al., 2022). As base models to fine-tune with our generated data, we use the vanilla LM Pythia-1.4B (Biderman et al., 2023) for ablation analysis, MPT-7B[2], a decoder only LM similar to LLaMA (Touvron et al., 2023) as well as GPT-JT-6B[3], an instructed version of GPT-J (Wang and Komatsuzaki, 2021) trained on Chain of Thought and Natural instruction datasets among others. All chosen models are open-source and have permissive licenses (Apache-2).

We evaluate the models fine-tuned on the data generated by Ensemble-Instruct on the the Super-Natural Instructions (SuperNI) test set (Wang et al., 2022) and 252 user-oriented tasks from Wang et al. (2023). Our contributions can be summarized as follows:

- We propose a technique for generating high-quality instruction-tuning data with 40B-parameter or smaller LMs that are openly accessible, with non-restrictive licenses.

- We outperform Self-Instruct training of GPT3 (175B) with a far smaller base model (MPT-7B). The technique also improves the performance of instruction-tuned GPT-JT-6B.

- Ablation studies demonstrate the importance of the individual components of our technique.

- We release the synthetic instruction-tuning dataset of about 45k samples along with our ICL templates and codebase.

## 2  Ensemble-Instruct

[1] https://huggingface.co/togethercomputer/GPT-NeoXT-Chat-Base-20B
[2] https://www.mosaicml.com/blog/mpt-7b
[3] https://huggingface.co/togethercomputer/GPT-JT-6B-v1

---

**Algorithm 1** Output Ensembling

**Input**: LM outputs $o_1$, $o_2$, $o_3$; Threshold $t$
**Output**: Best output $o_{best}$

1: $o_{best} \leftarrow$ None
2: $Rs \leftarrow \phi$
3: **for** $(i, j)$ in $\{(1, 2), (1, 3), (2, 3)\}$ **do**
4: $\quad Rs \leftarrow Rs \cup \text{RougeL}(o_i, o_j)$
5: **end for**
6: **if** $\min(Rs) > t$ **then**
7: $\quad i, j \leftarrow \text{argmax}(Rs)$
8: $\quad o_{best} \leftarrow o_i$
9: **end if**
10: return $o_{best}$

---

A high-level overview of Ensemble-Instruct is given in Figure 1. The algorithm has three main components: (i) Categorization of tasks and their associated prompts, (ii) Generation of instructions followed by instances, where an instance comprises an input (optional) and an output, and (iii) Ensemble of outputs from multiple LMs.

### 2.1  Categorization of Tasks and Prompts

We divide the tasks, i.e. the instruction-tuning samples, into two categories: those where the instruction needs an input to be meaningful (type A) and those where it does not (type B). Examples of tasks from these two types can be seen in Figures 1 and 2. Among the seed tasks of Wang et al. (2023), 125 belong to type A and 50 to type B. For each category, we employ a dedicated pipeline that (a) uses ICL demonstrations only of that type, and (b) tailors the number of demonstrations to the difficulty of the type, at different stages of generation.

### 2.2  Instruction Generation

For type A tasks, we use 24 ICL demonstrations during instruction generation. Out of those, 20 are randomly sampled from the 125 seed tasks of the same type, and 4 are sampled from instructions previously generated by the model itself. For type B tasks, we use 10 ICL demonstrations, of which 8 are sampled from the 50 type B seed tasks and 2 from previously generated synthetic instructions. Further, we adopt the approach of Wang et al. (2023) of adding a new instruction to the set only if its Rouge-L (Lin, 2004) score with every existing instruction is less than 0.7.

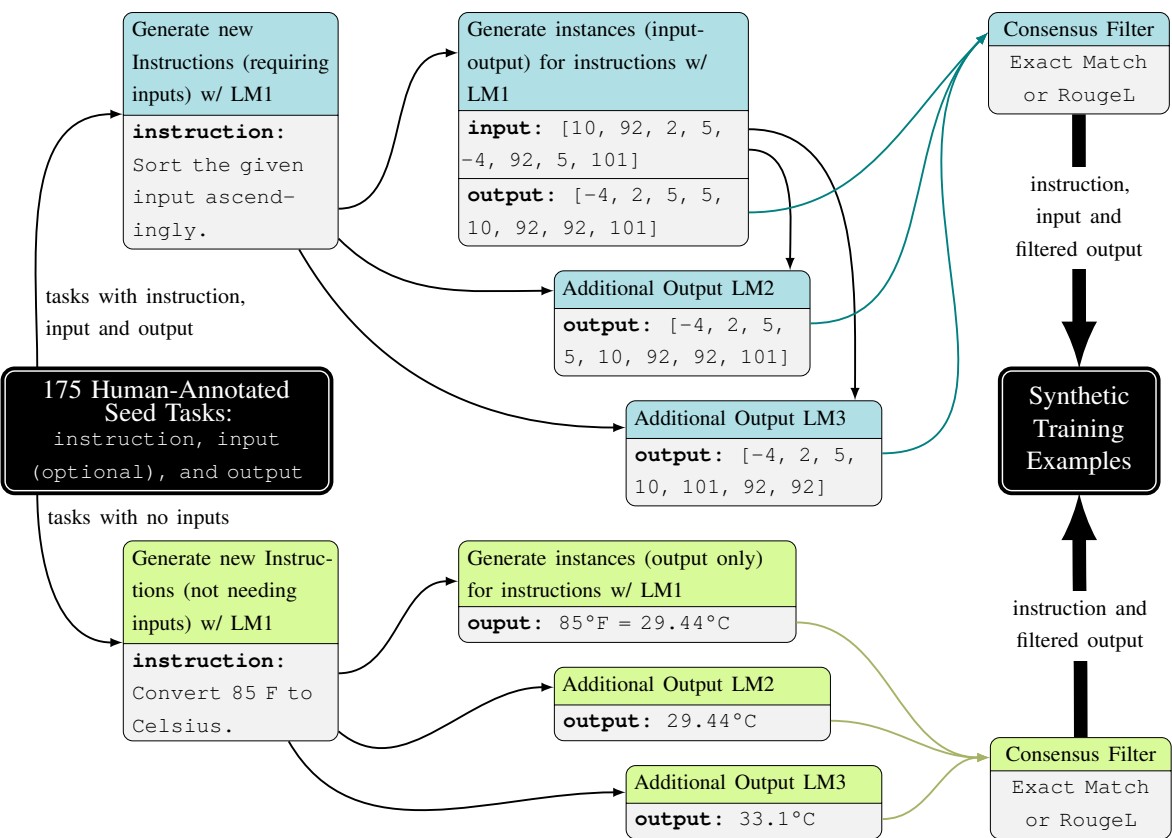

Figure 1: High-level overview of Ensemble-Instruct for synthetic instruction data generation. The top part generates data for the tasks comprising instruction, input, and output while the bottom part generates for tasks without inputs. The instruction generation and instance generation steps are done using the same LM with few-shot in-context learning. Additional LMs are used for the additional output generation, for which in-context examples are used only when the LM is not previously instruction tuned. In each box, the bottom gray portion gives an example of what is produced during that step.

## 2.3 Instance Generation

During instance generation, we use 18 ICL demonstrations for type A tasks and 15 for type B tasks, randomly selected from the seed tasks. Figure 2 shows examples of type A and type B tasks, and the prompts used for instance generation.

## 2.4 Output Ensembling

The instruction and instance generation steps should in principle complete the process of synthesizing an instruction-tuning sample (Wang et al., 2023). However, samples generated by small LMs can be inaccurate, which prompts us to design a final step of output ensembling. Instead of simply accepting the already generated example, we use an additional set of LMs to predict new outputs, given either the generated instruction-input pair (type A) or the instruction (type B).

The final output is derived by applying the greedy consensus Algorithm 1 to the outputs gener-

ated by the different LMs. The algorithm computes the Rouge-L score between all three pairs of outputs. If the lowest Rouge-L is above a threshold $t$, it returns the first element of the pair with the highest Rouge-L score. This can be seen as a greedy version of Minimum Bayesian Risk decoding (Goel and Byrne, 2000) with additional thresholding. The minimum threshold $t$ is set to 0.01 across all tasks. It is important to note that if the above process does not select any of the three outputs, the example is filtered out.

## 3 Analysis of Instruction Tuning Dataset

We generate multiple instruction-tuning datasets using a heterogeneous set of LMs. Table 1 shows the labels of our synthetic datasets according to the LMs used in different stages of generation. Table 2 summarizes the set of LMs we use for generation.

| Label | Instructions | Instances | Additional Outputs for Ensembling |
|---|---|---|---|
| SO-FALCON | FALCON | FALCON | – |
| SO-{UL2, NEOX} | UL2, GPT-NEOXT-CHAT | UL2, GPT-NEOXT-CHAT | – |
| EO-FALCON-LM | FALCON | FALCON | UL2, FALCON |
| EO-FALCON-ILM | FALCON | FALCON | FLAN-UL2, GPT-NEOXT-CHAT |
| EO-{UL2, NEOX}-ILM | UL2, GPT-NEOXT-CHAT | UL2, GPT-NEOXT-CHAT | FLAN-UL2, FLAN-T5-XXL |

Table 1: Labels of our synthetic tuning datasets according to the LMs used for generating instructions, instances and additional outputs for ensembling. Datasets with outputs from a single LM and an ensemble of LMs are prefixed with SO- and EO-, respectively. The rest of each label specifies the models that were used at different stages of the process. If additional outputs were generated using instruction-tuned LMs for ensembling, the dataset is suffixed with -ILM. If vanilla LMs were used for the same purpose, we use the suffix -LM. With instruction-tuned LMs, we generate the output zero-shot; for vanilla LMs, we use few-shot ICL.

## 3.1 Instance vs. Output Generation

As shown in Table 1, we use a distinct set of LMs for instruction and instance generation on one hand and output generation for ensembling on the other. The motivations are two-fold: (1) We observed that only relatively large decoder only models with 20B parameters or more are capable of generating input-output instances (type A). Therefore, we use decoder only models including FALCON, GPT-NEOXT-CHAT for input-output instance generation. (2) Instruction-tuned models are capable of generating high quality zero-shot outputs. Therefore, we use instruction-tuned models including FLAN-UL2, FLAN-T5-XXL, GPT-NEOXT-CHAT for additional output generation for ensembling. We found that vanilla LMs UL2, FALCON lag behind instruction-tuned models for output generation, as shown in EO-FALCON-LM of Table 4.

Table 3 reports the number of valid instance generations, as well as samples accepted by the ensemble Algorithm 1, using FLAN-UL2 and FLAN-T5-XXL as additional outputs. We show results for 100 random samples using different models (FALCON, FLAN-UL2, GPT-NEOXT-CHAT) to generate instruction and type A instances using the same prompt

| Model | # params | LM type | Rouge-L |
|---|---|---|---|
| FALCON | 40B | causal | 12.7 |
| UL2 | 20B | seq2seq | 10.4 |
| GPT-NEOXT-CHAT | 20B | causal[†] | 6.6 |
| FLAN-UL2 | 20B | seq2seq[†] | 77.5 |
| FLAN-T5-XXL | 11B | seq2seq[†] | 73.0 |

Table 2: LMs we used for instruction-tuning data generation. *seq2seq* denotes sequence-to-sequence and *causal* denotes decoder-only. GPT-NEOXT-CHAT is tuned on the OIG dataset[4]. FLAN-UL2 and FLAN-T5-XXL are tuned on FLAN collections. Both OIG and FLAN include SUPERNI data. Instruction-tuned models are denoted by [†]. Zero-shot performance of each model on the SUPERNI test set is provided in Rouge-L.

| Model | instruction | instance | ensemble |
|---|---|---|---|
| FALCON | 100 | 72 | 49 (68%) |
| GPT-NEOXT-CHAT | 100 | 40 | 25 (63%) |
| FLAN-UL2 | 100 | 0 | 0 (0%) |

Table 3: Number of valid type A instructions and instances generated by different models for 100 samples as well and number (and percentage) of samples filtered by Algorithm 1. All models share the same prompt and examples.

and examples [5]. Instructed models struggle to generate valid instances and in particular FLAN-UL2 generates no valid instance for the 100 samples. Although not shown in the table, most LMs are capable of generating type B instructions and instances, indicating that instructions and instances that do not require an input is an easier task than those requiring an input.

## 3.2 Small LM Dataset Comparsion

We instruction-tune Pythia-1.4B-deduped with different datasets and evaluate them on the 119 tasks of the SUPERNI test set. For validation, we use 10,589 samples from 106 SUPERNI training tasks. Note that the validation and test sets have zero task overlap. We instruction-tune the model for 5 to 7 epochs and select the checkpoint with the highest validation Rouge-L score for evaluation. Performances of these tuned models on the test set are shown in Table 4, where M-SELF-INST denotes the algorithm and ICL templates of Wang et al. (2023) applied to {UL2, NEOX}, and F-SELF-INST, the algorithm and ICL templates of Wang et al. (2023) applied to FALCON. We also show the performance of PYTHIA-1.4B-DEDUPED fine-tuned with two ex-

---
[5]See https://github.com/IBM/ensemble-instruct/blob/main/ensemble_instruct/sample_instances.py for instance rejection criteria and scripts/ensemble_instruct.sh for experiment reproduction.

| Dataset | # samples | Rouge-L |
|---|---|---|
| ZERO-SHOT BASELINE | 0 | 9.8 |
| ALPACA | 51,760 | 33.4 |
| SELF-INST | 82,612 | 34.4 |
| M-SELF-INST | 24,984 | 28.5 |
| SO-{UL2, NEOX} | 25,660 | 33.6 |
| EO-{UL2, NEOX}-ILM | 18,218 | 38.3 |
| F-SELF-INST | 38,624 | 25.6 |
| SO-FALCON | 30,537 | 34.4 |
| EO-FALCON-LM | 26,503 | 32.9 |
| EO-FALCON-ILM | 26,701 | 37.1 |

Table 4: Efficacy of synthetic instruction tuning datasets measured by the performance of PYTHIA-1.4B-DEDUPED tuned models on the SUPERNI test set. Dataset labels are described in Table 1. ALPACA and SELF-INST are external synthetic datasets for further comparisons. M-SELF-INST denotes the algorithm and ICL templates of Wang et al. (2023) applied to {UL2, NEOX}. F-SELF-INST denotes the algorithm and ICL templates of Wang et al. (2023) applied to FALCON. All training sets include the 175 seed tasks and the learning rate is 1e-5.

ternal datasets, ALPACA[6] and SELF-INST[7] for comparisons with much larger training data obtained with the SELF-INSTRUCT algorithm.

The performance gap between M-SELF-INST and SO-{UL2, NEOX} shows that our categorization and simplification of ICL prompts for instruction and instance generation already improves performance over Self-Instruct. The same applies to the larger FALCON model, with SO-FALCON outperforming F-SELF-INST by a large margin. Output ensembling with instruction-tuned LMs further improves performance in both settings. Importantly, we find ensembling with vanilla LMs via ICL less effective than ensembling with instruction-tuned LMs that were applied zero-shot. Finally, we produce data that is more sample-efficient than Self-Instruct: With only about 30k examples, SO-FALCON yields a Rouge-L score of 34.4, which is equal to what Self-Instruct yields with about 82k examples.

## 3.3 Qualitative Analysis

We randomly select 140 samples (40 with an input and 100 with no input) from EO-{UL2, NEOX}-

[6] https://huggingface.co/datasets/yahma/alpaca-cleaned
[7] https://github.com/yizhongw/self-instruct/blob/main/data/gpt3_generations/batch_221203/all_instances_82K.jsonl

| criteria | Instance Type | | |
|---|---|---|---|
| | output | input-output | total |
| GOOD | 77 | 22 | 99 (70.7%) |
| BAD | 14 | 15 | 29 (20.7%) |
| MAYBE | 9 | 3 | 12 (8.6%) |
| **total** | 100 | 40 | 140 |

Table 5: Manual evaluation of synthetic instruction tuning data quality on 140 randomly selected samples.

ILM and manually assign one of three categories to each: GOOD, BAD and MAYBE. GOOD indicates that there are no errors in the instruction, input (optional) and output, and the sample as a whole is coherent. MAYBE indicates that the input and the output do not contain errors, but the quality is questionable, e.g., the output is not complete. BAD indicates that the input or the output contains errors and is incoherent with the instruction.

Manual evaluation results are shown in Table 5, which was carried out by one of the authors. We find that examples containing only an instruction and an output (type B) are generally of higher quality (77% GOOD) than those also containing an input (type A) (55% GOOD). This difference in quality is reflective of the relative difficulty of generating them by smaller models, i.e. it is easier to generate output-only instances, as suggested in §3.1. Out of the 24,809 M-SELF-INST examples in Table 4 (after excluding the 175 seed tasks), 20,752 (83.6%) are of type B, further demonstrating that it is easier to generate output-only instances. Ensemble-Instruct pipeline avoids such unbalanced generation by first categorizing the tasks and then leveraging separate sets of simplified prompts for each. Each of our data sets generated with Ensemble-Instruct is an almost even split between instructions with and without an input.

Figure 3 shows some synthetic examples before and after output ensembling, depicting a few different ways in which ensembling improves the quality of the generated output. Regarding the effect of ensembling, observations show that it is particularly effective in selecting accurate output when it is short, e.g. classification tasks, via exact match. For longer outputs from generation tasks, e.g. summarization, the algorithm often filters out non-sensical outputs with hallucinations.

---

***Instance Generation with Both an Input and an Output:***

```
Generate examples for the following instructions. The instruction requires input and output
instances. And you have to generate both input and output.

instruction: Extract all the country names in the paragraph, list them separated by commas.
input: Dr. No is the sixth novel by the English author Ian Fleming to feature his British Secret
Service agent James Bond. Written at Fleming's Goldeneye estate in Jamaica, it was first published
in the United Kingdom by Jonathan Cape in 1958. In the novel Bond looks into the disappearance in
Jamaica of two fellow MI6 operatives who had been investigating Doctor No. Bond travels to No's
Caribbean island and meets Honeychile Rider, who is there to collect shells. They are captured and
taken to a luxurious facility carved into a mountain. The character of Doctor No, the son of a
German missionary and a Chinese woman, was influenced by Sax Rohmer's Fu Manchu stories. Dr. No was
the first of Fleming's novels to face widespread negative reviews in Britain, but it was received
more favourably in the United States.
output: English, British, Jamaica, the United Kingdom, German, Chinese, Britain, the United States.
|EoS|

instruction: Sort the given input ascendingly.
input: [10, 92, 2, 5, -4, 92, 5, 101]
output: [-4, 2, 5, 5, 10, 92, 92, 101]
|EoS|

instruction: In this task, you need to compare the meaning of the two sentences and tell if they
are the same. Output yes or no.
input: Sentence 1: The teacher is speaking to the class. Sentence 2: The teacher is
speaking to the students.
output: yes
|EoS|

instruction:
```

---

***Instance Generation with Only an Output:***

```
Generate examples for the instructions. The instruction does not require input and generate the
output directly.

instruction: Which exercises are best for reducing belly fat at home?
output: Lying Leg Raises, Leg In And Out, Plank, Side Plank, Sit-ups
|EoS|

instruction: Is working part-time a good idea for teenagers between 15 and 19 years old?
output: There are both pros and cons to having a part-time job as a teenager. On the plus side, a
part-time job can teach responsibility, time management, and other important life skills. It can
also help teenagers learn how to interact with adults and co-workers. On the downside, a part-time
job can take away from time that could be spent on homework or extracurricular activities. It can
also be difficult to find a job that works around a school schedule.
|EoS|

instruction:
```

---

Figure 2: Example ICL templates for instance generation. The top 3 examples are for instances with both an input and an output; the bottom 3 are for output-only instances. We append the special token |EoS| to the end of each example, clearly demarcating example boundaries, to make sure generation stops after the final output token.

## 4 Experimental Results

We carry out automatic evaluations using Rouge-L on the evaluation sets in Table 6. Both 119 SU-PERNI tasks and 252 user-oriented tasks are identical to those evaluated in Wang et al. (2023).[8]

| Evaluation Data | # tasks | # samples |
|---|---|---|
| SUPERNI | 119 | 11,810 |
| User-Oriented | 252 | 252 |

Table 6: Evaluation datasets for automatic evaluations using Rouge-L. None of the tasks in the evaluation are seen during training.

---

[8]We refer the reader to §4.4 of Wang et al. (2023) on the characteristics of the 252 user-oriented test set and §A.1 on

```
instruction: Find the maximum number of the number in a set of numbers.
input: 1, 2, 23, 50, 1, 2, 23, 50, 1, 6, 22
output-before: 1, 2, 23, 50, 1, 2, 23, 23
output-after: 50

instruction: Describe a specific feature of the product in one sentence.
input: The feature of our product is that it has a rounded body shape, which is convenient to carry around.
output-before: Our product has a round design, making it lightweight and portable.
output-after: The product has a rounded body shape.

instruction: You are given a description of an item. Your task is to predict whether it can be found in a
store or not. Your output should be yes or no.
input: a red rubber ball
output-before: no
output-after: yes

instruction: Please describe a situation where you felt really happy.
output-before: When you have a happy life and you are happy, you are happy for yourself. When you're happy
for yourself, you are happy for everybody.
output-after: I felt really happy when I won the game

instruction: What are some foods that are high in Vitamin C?
output-before: 1. Tomatoes 2. Broccoli 3. Strawberries 4. Papaya 5. Oranges
output-after: oranges
```

Figure 3: Instruction tuning dataset examples before and after output ensembling. Ensembling generally improves different aspects of output quality, including correctness and adherence to the specifics of the question. We observe a side effect of shorter outputs being preferred over longer ones in generation tasks even if in some cases that makes the output less accurate, as shown in the last example.

We set aside 106 tasks (10,589 samples) from the SuperNI 756 training tasks as the validation data set. For SuperNI instruction tuning, we exclude the validation set from training to simulate evaluation on unseen tasks.

We fine-tune 2 base LMs on the instruction tuning data generated by the current technique: (1) a vanilla LM, MPT-7B, and (2) an instruction tuned LM, GPT-JT-6B.[9] To fine-tune these models, we adopt QLoRA (Dettmers et al., 2023), which enables us to train both LMs with a single A100 GPU (40GB memory) within 24 hours. We also carried out full fine-tuning of MPT-7B for 2 data sets, EO-{UL2,NEOX}-ILM and SuperNI with 2 A100 GPUs (80GB memory). The results are shown in Tables 7 and 8 for the SuperNI test set, and in Table 9 for the 252 user-oriented test set.

In Table 7, MPT-7B fine-tuned on our synthetic data generated from vanilla LMs (SD I) outperforms both T0 and GPT3_{SELF-INST} despite the fact that the latter are fine-tuned on over 80K sam-

the analysis of the overlap between 175 seed instructions and the two evaluation data sets.

[9]They first train 2.62 billion tokens using the UL2 loss on the Pile, (Gao et al., 2020), followed by 0.92 billion tokens with a mixture of 5% of Chain-of-Thought (COT, Longpre et al. (2023)), 20% of Public Pool of Prompts (P3, (Bach et al., 2022)), 20% of SuperNI, and 55% of the Pile.

ples whereas MPT-7B is fine-tuned only on around 30K samples. MPT-7B fine-tuned on our synthetic data generated from instruction-tuned models (SD II) outperform the data generated using vanilla LMs (SD I) by up to 3 points. Full fine-tuning outperforms QLoRA fine-tuning by 1.4 on EO-{UL2,NEOX}-ILM (46.8 vs. 45.4). Full fine-tuning again outperforms QLoRA fine-tuning by 2.2 on SuperNI training (50.4 vs. 48.2). MPT-7B fine-tuned on the combination of two synthetic data sets EO-{UL2,NEOX ∪ FALCON}-ILM and the SuperNI training set improves the Rouge-L score over SuperNI training only by 2.2 points (from 48.2 to 50.4). We see a similar pattern in Table 8 for the instruction-tuned base LM GPT-JT-6B. The fact that our synthetically generated data significantly improve the performance of the instruction-tuned LM suggests that our technique generates data sufficiently different from the instruction tuning data incorporated into the base LM training.

In Table 9, we note that both base models, MPT-7B and GPT-JT-6B, perform worse on the user-oriented data set than on the SuperNI test set: 10.6 vs. 16.6 with MPT-7B and 6.2 vs. 10.4 with GPT-JT-6B. Fine-tuning these models on about 45K samples of the synthetic data provides a significant

| Models | # Params | Training Set | # Samples | Rouge-L |
|---|---|---|---|---|
| **Vanilla Base LMs** | | | | |
| T5-LM, Wang et al. (2023) | 11B | None (ZERO-SHOT) | 0 | 25.7 |
| GPT3, Wang et al. (2023) | 175B | None (ZERO-SHOT) | 0 | 6.8 |
| MPT | 7B | None (ZERO-SHOT) | 0 | 16.6 |
| **Instruction-tuned w/ SD I** | | | | |
| T0, Wang et al. (2023) | 11B | Self-Instruct (GPT3) | 82,612 | 33.1 |
| GPT3$_{\text{SELF-INST}}$, Wang et al. (2023) | 175B | Self-Instruct (GPT3) | 82,612 | 39.9 |
| MPT$^{\text{qlora}}$, ours | 7B | SO-FALCON | 30,537 | 43.1 |
| MPT$^{\text{qlora}}$, ours | 7B | EO-FALCON-LM | 26,503 | 43.2 |
| **Instruction-tuned w/ SD II** | | | | |
| MPT$^{\text{qlora}}$, ours | 7B | EO-FALCON-ILM | 26,701 | 44.4 |
| MPT$^{\text{ff}}$, ours | 7B | EO-{UL2,NEOX}-ILM | 18,218 | 46.8 |
| MPT$^{\text{qlora}}$, ours | 7B | EO-{UL2,NEOX}-ILM | 18,218 | 45.4 |
| MPT$^{\text{qlora}}$, ours | 7B | EO-{UL2,NEOX ∪ FALCON}-ILM | 44,744 | 46.4 |
| **Instruction-tuned w/ SUPERNI** | | | | |
| T$_k$-Instruct, Wang et al. (2023) | 11B | SUPERNI | 50,000 | 46.0 |
| GPT3, Wang et al. (2023) | 175B | SUPERNI | 50,000 | 49.5 |
| MPT$^{\text{ff}}$, ours | 7B | SUPERNI | 64,528 | 50.4 |
| MPT$^{\text{qlora}}$, ours | 7B | SUPERNI | 64,528 | 48.2 |
| **Instruction-tuned with SD II & SUPERNI** | | | | |
| GPT3$_{\text{SELF-INST}}$, Wang et al. (2023) | 175B | Self-Instruct & SUPERNI | 132,612 | 51.6 |
| MPT$^{\text{qlora}}$, ours | 7B | EO-COMBO-ILM & SUPERNI | 109,272 | 50.4 |

Table 7: Evaluation results on the SuperNI test set. SD I denotes synthetic data generated from only vanilla LMs, and SD II, synthetic data generated from the combination of vanilla and instruction-tuned LMs. Superscript$^{\text{ff}}$ denotes full fine-tuning. Superscript$^{\text{qlora}}$, QLoRA fine-tuning. Learning rate is set to 1e-6 for full fine-tuning and 5e-5 for QLoRA tuning. EO-COMBO-ILM denotes EO-{UL2, NEOX ∪ FALCON}-ILM. Combination of synthetic data EO-COMBO-ILM and SUPERNI training set improves over SUPERNI training set by 2.2 points, from 48.2 to 50.4. Instruction tuning with SD II output-performs instruction tuning with SD I. For instruction tuning with SuperNI, we subsample 100 instances from each of the 650 training tasks.

| Trainset | # Samples | Rouge-L |
|---|---|---|
| ZERO-SHOT | 0 | 10.4 |
| FALCON | 30,537 | 41.7 |
| EO-FALCON-LM | 26,503 | 40.5 |
| EO-FALCON-ILM | 26,701 | 41.9 |
| EO-{UL2,NEOX}-ILM | 18,218 | 42.7 |
| EO-COMBO-ILM | 44,744 | 43.1 |
| SUPERNI | 64,528 | 44.2 |

Table 8: Results of (instruction-tuned base LM) GPT-JT-6B fine-tuned on synthetic data. EO-COMBO-ILM denotes EO-{UL2, NEOX ∪ FALCON}-ILM. All models are fine-tuned with QLoRA with learning rate 5e-5.

| Models | Trainset | Rouge-L |
|---|---|---|
| MPT-7B | ZERO-SHOT | 10.6 |
| MPT-7B | M-SELF-INST | 20.6 |
| MPT-7B | F-SELF-INST | 21.6 |
| MPT-7B | EO-COMBO-ILM | 22.1 |
| GPT-JT-6B | ZERO-SHOT | 6.2 |
| GPT-JT-6B | M-SELF-INST | 16.5 |
| GPT-JT-6B | F-SELF-INST | 17.4 |
| GPT-JT-6B | EO-COMBO-ILM | 21.5 |

Table 9: Results on the 252 user-oriented test set.

boost to the Rouge-L scores, from 10.6 to 22.1 for MPT-7B, and from 6.2 to 21.5 for GPT-JT-6B. This suggests that the synthetic data we generate capture the characteristics of user-oriented instructions to a certain degree. Consistent with the results noted in Table 4 for the SuperNI test set, the data generated by our technique is more effective than the data generated using Self-Instruct (M-SELF-INST, F-SELF-INST) on the user oriented data set as well.

In Table 10, we show experimental results with other much larger models to illustrate the scalability of the proposed Ensemble-Instruct to any black-box models. Regardless of the base model sizes, ranging from 6B to 40B, fine-tuning the base model with the synthetic data EO-{UL2, NEOX ∪ FALCON}-ILM improves the Rouge-L score significantly. The fine-tuned model performances seem to correlate well with the base model's parameter sizes, i.e. 43.1 for the smallest GPT-JT-6B, 49.9 for the largest FALCON-40B and all other model sizes and scores in between. In particular, the experimental results on FALCON-40B indicates that Ensemble-Instruct is not an instance of model distillation in the sense that the synthetic data generated from FALCON-40B and smaller models signifi-

| Model-ParamSize | zero-shot | fine-tuned |
|---|---|---|
| GPT-JT-6B | 10.4 | 43.1 |
| MPT-7B | 16.6 | 46.4 |
| OPEN-LLAMA-13B | 11.9 | 46.7 |
| MPT-30B | 12.2 | 49.5 |
| FALCON-40B | 12.7 | 49.9 |

Table 10: Fine-tuning results on large models demonstrating the scalability of the Ensemble-Instruct technique to any black-box models. Zero-shot and fine-tuned model scores are Rouge-L on SUPERNI test set. Performance improvement of FALCON-40B after fine-tuning, compared with its zero-shot performance indicates that Ensemble-Instruct is not an instance of model distillation. All models are fine-tuned with EO-{UL2, NEOX ∪ FALCON}-ILM in Table 7.

cantly improves all model's zero-shot performance including the largest model FALCON-40B.

## 5 Related Work

This work is directly related to Self-Instruct (Wang et al., 2023), borrowing from it the initial seed tasks and the idea of using ICL for tuning a base model into a instruction following model. It could also be seen as related to follow-up works such as: Alpaca (Taori et al., 2023)—a practical application of Self-Instruct—Evol-Instruct (Xu et al., 2023), which iteratively evolves instructions into increasing difficulty levels and Dromedary (Sun et al., 2023b), which combines self-instruct with principle-based correction, similar to Constitutional AI (Bai et al., 2022). One fundamental limitation of these approaches is that they resort to very large language models (around 175B parameters or 65B parameters at the minimum) that are also proprietary and non-public. Here we explore techniques for generating instruction tuning data using LMs that are much smaller (around 10B–40B parameters) and have permissive licenses. We crucially draw on a heterogeneous mixture of smaller LMs to generate diverse outputs and then ensemble over multiple outputs to select high-quality synthetic examples, while also simplifying the instruction creation process.

The use of a reference metric, such as Rouge-L, to ensemble the outputs of multiple language distributions is a common technique in Minimum Bayesian Risk decoding, with applications to speech-to-text (Goel and Byrne, 2000), machine translation (Kumar and Byrne, 2004), language modeling (Suzgun et al., 2022) and parsing (Lee

et al., 2022), among others. Here we use a similar technique in the context of instruction generation. To the best of our knowledge, this is the first application of such an approach to instruction-tuning data generation.

Jiang et al. (2023) proposes LLM-Blender, an ensembling framework to improve the generaion qualities by leveraging the diverse strengths of multiple language models. While we utilize the output ensemble in the context of synthetic data generation with Rouge-L as the reference metric, LLM-Blender focuses on improving model output qualities using PairRanker and GenFuser, both approaches capitalize on the efficacy of ensembling as a way of improving output qualities.

Also related to this work are approaches directly distilling from ChatGPT or GPT-4 (OpenAI, 2023) without specific instruction strategies, such as Vicuna[10], which distills ChatGPT, Baize (Xu et al., 2032), distilling conversations and Orca (Mukherjee et al., 2023), which uses a large amount of ChatGPT and GPT-4 outputs and combines FLAN tasks, system prompts and machine-generated explanations sampled from these models. The strength of these approaches seems to rely more on the amount and quality of teacher samples available than on the inductive biases of the self-instructing technique and still rely on proprietary models with non-permissive licenses.

## 6 Conclusion

We present a novel technique to generate instruction-tuning data through ICL, following the recent Self-Instruct work (Wang et al., 2023). Unlike Self-Instruct, we propose techniques that explicitly avoid the use of proprietary language models like GTP-3, ChatGPT or GPT-4. We show that when using smaller models, Self-Instruct becomes less performant. To overcome this, we draw on two main ideas: (a) Categorization and simplification of ICL templates to make prompt learning easier, and (b) Ensembling over multiple LM outputs to select high-quality examples. These ideas allow us to outperform training with Self-Instruct while utilizing the same seed tasks. The resulting synthetic data enables base models like MPT-7B to outperform GPT-3, a far larger model with 175B parameters. The results of this work also encourage the departure from closed-access models for advancing instruction generation algorithms.

---

[10] https://lmsys.org/blog/2023-03-30-vicuna/

# 7 Limitations

Due to time and resource constraints, some parts of the experimental setup are not ideal. All model outputs were collected from an internal API serving models from HuggingFace[11]. Due to limitations of this API, different number of samples were collected for each model which may have introduced noise in the performance estimates. We report the exact number of samples used for training along with the results. Note that for cases using ensembling one has to take into account that there is an additional filtering process that removes samples. We provide approximate rates for ensembling filtering in Table 3. For the small user-oriented test set containing 252 tasks, automatic evaluation is arguably not ideal. Proper human evaluation would provide a clearer signal but this requires of significant time investment and resources. The method employs a set of various LMs, and therefore the generated synthetic data can be susceptible to the limitations of such LMs, particularly the biases inherent in the training data which may be harmful leading to synthetic data with hate, abuse and social stereotypes.

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
