# OpenReview forum: "Ensemble-Instruct: Instruction Tuning Data Generation with a Heterogeneous Mixture of LMs"
_EMNLP/2023/Conference — EMNLP 2023 Findings_

### Official Review · Reviewer_gP8F · 2023-08-05

**Soundness:** 3

**Excitement:**

3: Ambivalent: It has merits (e.g., it reports state-of-the-art results, the idea is nice), but there are key weaknesses (e.g., it describes incremental work), and it can significantly benefit from another round of revision. However, I won't object to accepting it if my co-reviewers champion it.

**Paper Topic And Main Contributions:**

The paper proposes a new methodology for creating instruction-tuning data using LLM.
Particularly, instead of using excessively large and less accessible 175B-level models that were commonly used in the past, it suggests utilizing relatively smaller models below 40B.
The proposed method consists of two key methodologies: the first one involves the categorization and simplification of ICL templates, and the second one is about ensembling multiple LM outputs
Through experiments, the proposed approach demonstrates superior data generation compared to self-instruct, shows good performance on both vanilla and ILM, and indicates that smaller ILMs generate data better than larger LMs.

**Questions For The Authors:**

- A: Why do UL2 and NEOX appear together? Specifically, isn't UL2 vanilla model and NEOX ILM?
- B: Is the generation via sampling? In that case, what about generating multiple SOs and self-ensembling them?
- C: Why do you think the vanilla model ensembling is not very effective?

**Reasons To Accept:**

- The simplicity of the methodology is a key advantage, as it facilitates easy application, which is commendable.
- Despite utilizing smaller models, the approach achieves notable performance, particularly demonstrating improved results even with a limited number of samples.
- The insightful analysis of the differences between ILM and LM, following experimentation with both models, adds valuable understanding to the study.

**Reasons To Reject:**

- The proposed methodology appears disconnected from the primary motivation of the paper, which revolves around the model's size and openness. Although it was demonstrated using small models, there is potential for its applicability to all black-box models. To better showcase the effectiveness of the methodology, conducting experiments with larger models would be beneficial.

- The analysis of the effectiveness of categorization and simplification lacks in-depth exploration, and it would be advantageous to delve further into this aspect. While section 3.2 briefly mentions the balance between type A and B, a more comprehensive analysis would add valuable insights.

- The absence of a comparison with other datasets, such as Alpaca mentioned in the paper, is somewhat regrettable. Including such comparisons would provide a more comprehensive evaluation of the proposed approach. Additionally, a qualitative comparison with other methods would be valuable in further validating the proposed methodology.

**Reproducibility:**

4: Could mostly reproduce the results, but there may be some variation because of sample variance or minor variations in their interpretation of the protocol or method.

**Reviewer Confidence:**

4: Quite sure. I tried to check the important points carefully. It's unlikely, though conceivable, that I missed something that should affect my ratings.

**Typos Grammar Style And Presentation Improvements:**

- While there are citations, it would greatly enhance the paper's readability and understanding if it directly compares itself to Self-instruct, highlighting the differences.
- In T7, there seems to be a reference to SO-FALCON, but the "SO" part is missing, which causes some confusion.
- It would be beneficial to move the examples to the appendix and use the main content to provide additional analysis and supplementary information. This adjustment would make the paper even better, as the current amount of space dedicated to examples feels excessive relative to the information they provide.

---

> ### Author Rebuttal · Authors · 2023-08-24
>
> Thank you for your insightful and constructive review.
>
> **Rebuttals to "Reasons to Reject"**
>
> **Critic**: The proposed methodology appears disconnected from the primary motivation of the paper, which revolves around the model's size and openness. Although it was demonstrated using small models, there is potential for its applicability to all black-box models. To better showcase the effectiveness of the methodology, conducting experiments with larger models would be beneficial.
>
> **Response**: We provide new  experimental results on three larger models, including falcon-40b, mpt-30b and open_llama-13b. All models are qlora fine-tuned with 44,744 synthetic samples generated by our ensemble-instruct technique, i.e. EO-{UL2, NEOX U FALCON}-ILM in Table 6. Experimental results on the SNI evaluation data set are shown below, where the numbers in parentheses are obtained after post-processing the model outputs.
>
> | Models                                    |   zeroshot baseline RougeL |  qlora fine-tuned RougeL |
> | :--                                            |     --:                      | --:                       |
> | GPT-JT-6b (Table 6)                 |    10.4 (36.4)          |  43.1                   |
> | mosaicml/mpt-7b (Table 6)      |    16.6 (31.3)          |   46.4                  |
> | **open_llama-13b  (new)**       |    11.9 (32.6)          |  46.7                   |
> | **mosaicml/mpt-30b  (new)**   |    12.2 (44.8)         |  49.5                   |
> | **tiiuae/falcon-40b (new)**       |     12.7 (41.4)        |     49.9             |
>
> **Critic**: The analysis of the effectiveness of categorization and simplification lacks in-depth exploration, and it would be advantageous to delve further into this aspect. While section 3.2 briefly mentions the balance between type A and B, a more comprehensive analysis would add valuable insights.
>
> **Response**: Table 3 of Section 3.1 provides a quantitative analysis.  M-SELF-INST denotes the algorithm and ICL templates of SELF-INST applied to {UL2, NEOX} whereas SO-{UL2,NEOX} denotes our  algorithm incorporating categorized and simplified ICL templates. While RougeL of M-SELF-INST is 28.5, SO-{UL2,NEOX} achieves 33.6, i.e. 5.1 point improvement. F-SELF-INST denotes the algorithm and ICL templates of SELF-INST applied to FALCON whereas SO-FALCON denotes our algorithm incorporating categorized and simplified ICL templates. RougeL of F-SELF-INST is 25.6 compared with RougeL of SO-FALCON 34.4, i.e. 8.8 point improvement.
>
> **Critic**: The absence of a comparison with other datasets, such as Alpaca mentioned in the paper, is somewhat regrettable. Including such comparisons would provide a more comprehensive evaluation of the proposed approach. Additionally, a qualitative comparison with other methods would be valuable in further validating the proposed methodology.
>
> **Response**:  To  address this concern, we fine-tuned PYTHIA-1.4B-DEDUPED LM on https://huggingface.co/datasets/yahma/alpaca-cleaned under the same experimental condition as all other models in Table 3. Shown below is the summary, where Alpaca dataset performs similarly to SELF-INSTRUCT but worse than our proposed Ensemble-Instruct.
>
> | Dataset                                                |    Sample size    |  RougeL |
> | :--                                                         | :--                      | :--           |
> | **Alpaca (new)**                                  |   51,760            |    33.4    |
> | SLEF-INSTRUCT (Table  3)                 |   82,612             |    34.4     |
> | EO-{UL2, NEOX}-ILM (Table 3)           |   18,218             |   38.3      |
> | EO-FALCON-ILM (Table 3)                 |   26,701             |   37.1      |
>
> **Response to "Questions For The Authors":**
>
> **A**: Why do UL2 and NEOX appear together? Specifically, isn't UL2 vanilla model and NEOX ILM?
>
> **Response**: Before the release of falcon-40b (06/09/2023, 1 week before abstract submission deadline) and mpt-30b (06/28/2023) models, UL2 and NEOX (each 20b parameters) were the only models capable of generating both the input and the output given an instruction. Since we were looking for ways of diversifying the instructions (and consequently the input and output instances), we resorted to these models for initial instance generations.
>
> **B**: Is the generation via sampling? In that case, what about generating multiple SOs and self-ensembling them?
>
> **Response**:
>
> * For input-output generation given instruction, generation is done via sampling. For zero-shot output generation given instruction and  input, generation is done via greedy decoding.
>
> * We tried ensembling multiple outputs generated via sampling for falcon-40b, which turned out to be similar to or slightly lower than the performance of a single output, cf. EO-FALCON-LM in Table 3.
>
> **C**: Why do you think the vanilla model ensembling is not very effective?
>
> **Response**: For instruction tuned models such as flan-t5-xxl and flan-ul2, high quality outputs are generated zero-shot given the instruction and input.   On the other hand, output generation requires few shot learning with ICL templates for vanilla auto-regressive LMs. For paper submission, we used the same ICL templates for output generation as instance generation, which was not effective. However, our ongoing investigation indicates that vanilla model ensembling can also be effective with re-design of ICL templates and diverse outputs from multiple LMs, e.g. Falcon and LLaMA-2.

---

### Official Review · Reviewer_CwUv · 2023-08-05

**Soundness:** 4

**Excitement:**

4: Strong: This paper deepens the understanding of some phenomenon or lowers the barriers to an existing research direction.

**Paper Topic And Main Contributions:**

Authors propose a novel method of generating instruction tuning data, which can then be used to modify LLMs be become better aligned to human preferences.  They offer two key techniques (1) Categorizing and simplifying the ICL prompts and (2) ensembling over multiple LM outputs to improve accuracy and diversity. Ensemble-Instruct leads to a "small" model (MPT-7B) outperforming a much larger one (GPT 3 at 175B) trained using the typical Self-Instruct method, as tested on two benchmarks (SuperNI and User-oriented tasks)

Categorization is broken down into two types: (a) those with inputs and (b) those without inputs.  (The required parts of an instruction sample are the instruction and the output.)   By separating these types, the prompts can be simplified without losing any impact.  Specifically, Pipeline A (for inputs) includes adding the input during data generation.  In contrast, Pipeline B (without inputs) includes a different seed set of data.

The ensemble of multiple LMs (a) improves diversity by using many different types of LMs and (b) using majority vote of LMs to improve quality of data.  The voting is conducted by comparing the Rouge-L score of candidate generations, keeping only those above a certain threshold.

**Questions For The Authors:**

Can you please add a Figure to visually show the process of generating instruction-tuning samples?

Did you consider ensembling based on model confidence scores, rather than RougeL?

Is there a graph that shows how much downstream results improve as a you add more synthetic data?  I'd be curious to know if there is some tipping point where performance improves dramatically.  Also, include the point at which your ensemble-instruct data surpasses the self-instruct data.

**Reasons To Accept:**

Paper is well written and easy to follow.  Ensemble-Instruct technique shows clear gains over Self-Instruct across a wide variety of settings.

Ablation studies show that both aspects (simplified prompts, ensemble) are both useful in leading to superior performance.

Synthetic dataset of instruction-tuning examples can be used by others to build on.  The dataset only includes 45k examples, but I can see this being expanded to 100K+ examples if the codebase is opened up to others.

Anything that works with open-source, permissive licenses is great for the community, especially when doing so outperforms the black-box counterparts.

**Reasons To Reject:**

Ensembling is not a particularly novel way to denoise generated samples.  More sophisticated methods exists and it would have been nice to compare against them.

Could be improved with more experiments on how many candidates to ensemble, or how many examples to include during ICL.

Missing larger scale human evaluation for qualitative review.

**Reproducibility:**

4: Could mostly reproduce the results, but there may be some variation because of sample variance or minor variations in their interpretation of the protocol or method.

**Reviewer Confidence:**

3: Pretty sure, but there's a chance I missed something. Although I have a good feel for this area in general, I did not carefully check the paper's details, e.g., the math, experimental design, or novelty.

---

> ### Author Rebuttal · Authors · 2023-08-24
>
> Thank you for insightful and helpful review.
>
> **Rebuttals to "Reasons To Reject":**
>
> **Critic**: Ensembling is not a particularly novel way to denoise generated samples. More sophisticated methods exists and it would have been nice to compare against them.
>
> **Response**: We will try to explore techniques other than ensembling to denoise generated samples.
>
> **Critic**: Could be improved with more experiments on how many candidates to ensemble, or how many examples to include during ICL.
>
> **Response**:  We will try to incorporate the analysis in our next round of revisions.
>
> **Critic**: Missing larger scale human evaluation for qualitative review.
>
> **Response**: We originally planned human evaluation of the 252 user-oriented test set, which we did not get to due to various constraints. And human evaluations of generative model outputs take a high priority in our next things to do.
>
> **Responses to "Questions For The Authors":**
>
> **Q**: Can you please add a Figure to visually show the process of generating instruction-tuning samples?
>
> **Response**: Figure 1 is intended to be a visual illustration of our instruction tuning sample generation process.
>
> **Q**: Did you consider ensembling based on model confidence scores, rather than RougeL?
>
> **Response**: Although we thought of this possibility, it was not feasible due to our black box approaches to the language models for data generation. We are currently integrating the synthetic data generation pipeline into our model fine-tuning and inferencing framework, which will enable us to utilize model confidence scores in addition to external evaluation metrics for emsembling model outputs.
>
> **Q**: Is there a graph that shows how much downstream results improve as a you add more synthetic data? I'd be curious to know if there is some tipping point where performance improves dramatically. Also, include the point at which your ensemble-instruct data surpasses the self-instruct data.
>
> **Response**: We applied our ensemble-based approach to content-grounded synthetic data generation for RAG-style QA tasks and observed that adding more synthetic data to human annotated data improved response generation system performances. We will try to plot graphs, illustrating the impact of synthetic data on downstream tasks both in terms of synthetic data size and quality.

---

### Official Review · Reviewer_NX6N · 2023-08-13

**Typos Grammar Style And Presentation Improvements:** 1. It is desirable to briefly describ…
**Soundness:** 4

**Excitement:**

3: Ambivalent: It has merits (e.g., it reports state-of-the-art results, the idea is nice), but there are key weaknesses (e.g., it describes incremental work), and it can significantly benefit from another round of revision. However, I won't object to accepting it if my co-reviewers champion it.

**Missing References:**

n/a

**Paper Topic And Main Contributions:**

The paper proposes an improved ICL sample generation method based on several permissive LLMs. The main idea is rather simple: it treat ICL templates (i.e., tasks) with and without input separately; for each task, it uses a greedy consensus algorithm to select high-quality samples based on the output of multiple LLMs.



**Questions For The Authors:**

1. The current method is not really selecting "high-quality" samples, but more like selecting "popular" samples, which may further magnifies the latent biases in the training data due to the training set exposure to different LLMs.

2. Why separating the two types of tasks help?


**Reasons To Accept:**

* The paper proposes a feasible method to create high-quality ICL samples for small (and hence less-capable) LLMs. This topic has practical importance in many application areas.

* The evaluation is quite comprehensive, covering many settings and using several popular permissive LLMs.

**Reasons To Reject:**

* The main performance metric used is the Rouge-L score, which is far from being satisfactory. If human evaluation is too costly, at least some AI evaluation could be done (e.g., using GPT-4).

* The key step is the high-quality sample selection from multiple LLMs. Although these LLMs were built by different groups, there might still be some hidden correlation between their outputs (e.g., due to training on the same subset of data). Hence, the current method is not really selecting "high-quality" samples, but more like selecting "popular" samples, which may further magnifies the latent biases in the training data.

* There is no detailed case study, e.g., comparing the samples generate by the proposed method and those by the Self-Instruction method. Additional, such case studies can be carried out among different LLMs used in this study. It is desirable to see insightful analysis or trends from these studies.

**Reproducibility:**

5: Could easily reproduce the results.

**Reviewer Confidence:**

3: Pretty sure, but there's a chance I missed something. Although I have a good feel for this area in general, I did not carefully check the paper's details, e.g., the math, experimental design, or novelty.

---

> ### Author Rebuttal · Authors · 2023-08-27
>
> Thank you for the insightful and helpful review.
>
> **Rebuttals to Reasons To Reject:**
>
> **Critic**: The main performance metric used is the Rouge-L score, which is far from being satisfactory. If human evaluation is too costly, at least some AI evaluation could be done (e.g., using GPT-4).
>
> **Response**: We originally planned human evaluation of the 252 user-oriented test set, which we did not get to due to various constraints. Human evaluations of generative model outputs take a high priority in our next things to do.
>
> **Critic**: The key step is the high-quality sample selection from multiple LLMs. Although these LLMs were built by different groups, there might still be some hidden correlation between their outputs (e.g., due to training on the same subset of data). Hence, the current method is not really selecting "high-quality" samples, but more like selecting "popular" samples, which may further magnifies the latent biases in the training data.
>
> **Response**: We acknowledge that the current ensemble algorithm selects “popular samples” in the sense that  Algorithm I may be viewed as “majority voting”.
>
> **Critic**: There is no detailed case study, e.g., comparing the samples generate by the proposed method and those by the Self-Instruction method. Additional, such case studies can be carried out among different LLMs used in this study. It is desirable to see insightful analysis or trends from these studies.
>
> **Response**: In Section 3.2 (lines 239--247), we analyze that the examples generated by SELF-INST mostly consists of output-only instances (83.6%) whereas our proposed Ensemble-Instruct pipeline avoids such unbalanced generation. We will add more detailed analyses on the capabilities and limitations of different LLMs  in the final version of the paper.
>
> **Responses to "Questions For The Authors":**
>
> **Q**: The current method is not really selecting "high-quality" samples, but more like selecting "popular" samples, which may further magnifies the latent biases in the training data due to the training set exposure to different LLMs.
>
> **Response**: We acknowledge that the current ensemble algorithm selects “popular samples” in the sense that Algorithm I may be viewed as “majority voting”.
>
> **Q**: Why separating the two types of tasks help?
>
> **Response**: We initially observed that none of the LMs with 20b parameters or less was capable of generating decent input-output instances when ICL templates included the mix of output-only and input-output instances with heterogeneous labels for instruction, input and output. We hypothesized that separating out the two types of instances as well as unifying the labels would make the in-context learning easier for the LMs weaker than GPT3. And these hypothesis seems to be borne out. In Table 3 of Section 3.1, M-SELF-INST denotes the algorithm and ICL templates of SELF-INST applied to {UL2, NEOX} whereas SO-{UL2,NEOX} denotes our  algorithm incorporating categorized and simplified ICL templates. While RougeL of M-SELF-INST is 28.5, SO-{UL2,NEOX} achieves 33.6, i.e. 5.1 point improvement. F-SELF-INT denotes the algorithm and ICL templates of SELF-INST applied to FALCON whereas SO-FALCON denotes our algorithm incorporating categorized and simplified ICL templates. RougeL of F-SELF-INST is 25.6 compared with RougeL of SO-FALCON 34.4, i.e. 8.8 point improvement.

---

### Meta-Review · Area_Chair_Nofw · 2023-09-18

**Recommendation:** 4

**Metareview:**

Proposes a method to generate instruction-tuning data using smaller LLMs in contrast to most earlier work such as Self-Instruct. The main idea is to ensemble multiple LLMs via a greedy consensus algorithm (high inter-model ROUGE-L) to select high-quality samples; the result is similar performance to using much larger closed-LLMs, as measured by ROUGE-L. One issue with closed-LLMs is they often do not have permissive licenses and so a way to generate IT data without them is potentially valuable to the open-source community.

There were concerns about of using solely ROUGE-L as eval with no human evaluation. In particular, ROUGE-L is also used in the consensus algorithm. There is a risk of over optimizing on that metric.

---

### Decision · Program_Chairs · 2023-10-07

**Decision:**

Accept-Findings

**Comment:**

Proposes a method to generate instruction-tuning data using smaller LLMs in contrast to most earlier work such as Self-Instruct. The main idea is to ensemble multiple LLMs via a greedy consensus algorithm (high inter-model ROUGE-L) to select high-quality samples; the result is similar performance to using much larger closed-LLMs, as measured by ROUGE-L. One issue with closed-LLMs is they often do not have permissive licenses and so a way to generate IT data without them is potentially valuable to the open-source community.

There were concerns about of using solely ROUGE-L as eval with no human evaluation. In particular, ROUGE-L is also used in the consensus algorithm. There is a risk of over optimizing on that metric.